# OpenReview forum: "Refined generalization analysis of the Deep Ritz Method and Physics-Informed Neural Networks"
_ICML.cc/2025/Conference — ICML 2025 poster_

### Official Review · Reviewer_sHz9 · 2025-02-25

**Overall Recommendation:** 3

**Summary:**

The paper proposed refined generalization bounds for the Deep Ritz Method (DRM) and Physics-Informed Neural Networks (PINNs), including the Poisson equation and static Schrödinger equation on the $d$-dimensional unit hypercube with the Neumann boundary condition.

## update after rebuttal
Most of my concerns have been solved, so I maintain my positive rating.

**Claims And Evidence:**

Yes.

**Essential References Not Discussed:**

No.

**Experimental Designs Or Analyses:**

Not applicable. There are no experiments to validate the proposed theoretical claims.

**Methods And Evaluation Criteria:**

Not available.

**Other Comments Or Suggestions:**

No.

**Other Strengths And Weaknesses:**

1. Although the paper demonstrates good generalization performance in low-dimensional cases, how to extend these theories to more complex high-dimensional problems, especially when the solution of the PDE belongs to complex spaces, still requires further research.

2. The paper primarily focuses on Neumann boundary conditions and linear PDEs, with limited in-depth discussion of other types of boundary conditions (such as mixed boundary conditions) or nonlinear PDEs. Extending this work to more diverse physical problems is a potential research direction.

3. While the paper proposes improved generalization analysis and approximation rates, the consumption of computational resources and time remains an issue in practical applications, particularly for complex physical scenarios and large-scale problems. Further exploration is needed on how to reduce computational costs while maintaining accuracy.

4. Viewing PINNs as a multi-task learning problem is an effective framework, but it may lead to interference between tasks, especially when the nature of the tasks varies significantly. Further optimization of task balancing and interaction in multi-task learning could enhance the model’s performance.

**Questions For Authors:**

Please refer to the Other Strengths And Weaknesses section.

**Relation To Broader Scientific Literature:**

This paper proposed  refined generalization analysis for Deep Ritz Method (DRM) and Physics-Informed Neural Networks (PINNs).

**Theoretical Claims:**

The paper presents a logically structured set of theoretical claims supported by established mathematical techniques.

---

> ### Author Rebuttal · Authors · 2025-03-31
>
> We sincerely appreciate your time in reviewing our manuscript and your valuable insights. Below, we address your questions point by point.
>
> **Q1**: Extension of theories to more complex high-dimensional problems.
>
> **A1**: In this work, we analyze two scenarios: (1) when the solutions of PDEs reside in Barron spaces, and (2) when they belong to Sobolev spaces. For Sobolev spaces, certain constants in our results demonstrate exponential dependence on dimensionality, which limits their applicability to high-dimensional problems. This limitation is inevitable because the complexity of Sobolev spaces grows dramatically with increasing dimensions, leading to exponential scaling of certain constants in neural network approximation errors. In contrast, for Barron spaces, the constants in the generalization error exhibit at most polynomial dependence on dimensionality, making our results meaningful for high-dimensional settings.
>
> To establish fundamental theoretical insights, this work primarily analyzes the settings where solutions in Barron or Sobolev spaces. The extension to more complex spaces is an important direction for future research. We conjecture that our methodology can be generalized to these settings with appropriate modifications.
>
>
> **Q2**: Extension to other types of boundary conditions and nonlinear PDEs.
>
> **A2**: Regarding the extension to other boundary conditions, such as the following mixed boundary conditions:
> $$ -\Delta u+Vu =f \ in  \ \Omega, \ u+\beta\frac{\partial u}{\partial n}=g \ on   \ \partial \Omega,$$
> where $\beta\neq 0$. A direct analysis shows that the solution $u^{* }$ of this equation satisfies that
> $$u^{* }=\mathop{\arg\min} _{u \in H^1(\Omega) } \mathcal{E}(u):=\mathop{\arg\min} _{u \in H^1(\Omega) } \int _{\Omega} \left(||\nabla u||^2+Vu-2fu \right)dx+ \frac{1}{\beta} \int _{\partial \Omega} \left(u^2-2gu \right)ds. $$
> Moreover, for any $u\in H^1(\Omega)$, we have
> $$ \mathcal{E}(u)-\mathcal{E}(u^{* })\lesssim ||u-u^{* }|| _{H^1(\Omega)}^2\lesssim \mathcal{E}(u)-\mathcal{E}(u^{* }),$$
> which is similar to the strong convexity property required in our analysis for Neumann boundary conditions (see equation (4) and (6)), suggesting that our method can be naturally extended to mixed boundary conditions. For further details, please refer to Section D.2 (line 2851), where we discuss other boundary conditions.
>
> Regarding the extension to nonlinear PDEs within the PINNs framework, our method can also  generalize to a broad class of nonlinear PDEs. For instance, let us consider the following equation:
> $$\mathcal{D}(u)=f(u) \ in \ \Omega, \ u=g \ on   \ \partial \Omega,$$
> where $\mathcal{D}$ is a linear differential operator and $f$ is the Lipschitz nonlinear term. Here, for simplicity, we denote the loss function as
> $$L(u)=||\mathcal{D}(u)-f(u)|| _{L^2(\Omega)}^2+||u-g|| _{L^2(\partial \Omega)}^2.$$
> Assume that $u^{* }$ is the true solution, then we can deduce that
> \begin{align*}
> L(u)&=||\mathcal{D}(u)-f(u)-\mathcal{D}(u^{* })-f(u^{* })|| _{L^2(\Omega)}^2+||u-u^{* }|| _{L^2(\partial \Omega)}^2 \\\\
> &\lesssim ||\mathcal{D}(u)-\mathcal{D}(u^{* })|| _{L^2(\Omega)}^2 +||f(u)-f(u^{* })|| _{L^2(\Omega)}^2+ ||u-u^{* }|| _{H^1(\Omega)}^2\\\\
> &\lesssim ||\mathcal{D}(u)-\mathcal{D}(u^{* })|| _{L^2(\Omega)}^2 +||u-u^{* }|| _{L^2(\Omega)}^2+ ||u-u^{* }|| _{H^1(\Omega)}^2.
> \end{align*}
> Therefore, this bound is similar to that in the linear case, and our method remains applicable.
>
> Moreover, we agree with the reviewer's suggestion. Extending this framework to broader physical problems (e.g., inverse problems) is indeed an important direction.
>
> **Q3**: Practical applications.
>
> **A3**: In this work, we primarily focus on generalization analysis and approximation rates, while optimization aspects are beyond the scope of this study. As you rightly pointed out, the computational resource and time requirements of neural network-based PDE solvers indeed limit their applicability to complex physical systems and large-scale problems. Developing computationally efficient optimization methods for these solvers with guaranteed accuracy remains a key focus of our ongoing research.
>
> **Q4**: Issues in viewing PINNs as a multi-task learning  problem.
>
> **A4**: From a theoretical perspective, formulating PINNs as a multi-task learning problem allows us to derive tighter generalization bounds through the use of local Rademacher complexity in the multi-task setting. In the experimental perspective, as you mentioned, interactions between different tasks may indeed lead to certain issues, since the loss functions contain multiple different additive terms that
> can disagree and yield conflicting update directions. Some recent studies, such as [1] and [2], have developed efficient algorithms to mitigate such conflicts.
>
> **References**:
>
> [1]: Config: Towards conflict-free training of physics informed neural
> networks. ICLR 2025.
>
> [2]: Dual cone gradient descent for training physics-informed
> neural networks. NeurIPS 2024.

---

### Official Review · Reviewer_FWfm · 2025-03-11

**Overall Recommendation:** 3

**Summary:**

The manuscript presents a detailed error analysis for physics-informed neural networks (PINNs) and the deep Ritz method (DRM) for linear elliptic equations. Both, the case of the true solution belonging to a Barron space and a Sobolev space are discussed. Additionally, certain approximation theoretic results are provided, noteably approximation rates in Barron spaces are proven with explicit control over the neural networks weights, which is of independent interest. The manuscript is carefully written, well understandable and technically mature.

## update after rebuttal
I maintain my positive assessment of the manuscript.

**Claims And Evidence:**

All results are substantiated with complete proofs.

**Essential References Not Discussed:**

None that I am aware of.

**Experimental Designs Or Analyses:**

Not applicable.

**Methods And Evaluation Criteria:**

Not applicable.

**Other Comments Or Suggestions:**

The main concern I have with this paper is whether it fits the scope of ICML. It is a pure error analysis paper with no simulations. In my opinion, the manuscript would benefit from the review process in a mathematical journal focused on numerical analysis more than the review process at ICML. This concern does not influence my rating of the paper.

On page 8 of the manuscript, the authors discuss the relation of the PINN loss to the error measured in certain norms. There are estimates available in the literature for precisely this question, see https://academic.oup.com/imajna/advance-article-abstract/doi/10.1093/imanum/drae081/7904789

**Other Strengths And Weaknesses:**

Strengths:
- The analysis is -- up to the optimization error -- complete. The manuscript is the most comprehensive error analysis of the DRM and PINNs that I am aware of.
- It treats both the setting of Sobolev spaces and Barron spaces.

Weakness:
- The manuscript would benefit from simulations illustrating some of the theoretical findings. I think this is essential and will add value to the paper. From a practitioners point view, it is important to know to which extent theory and practice meet.
- From a practical/optimization standpoint, it is more natural for the deep Ritz case to consider $H^2$ conforming networks. While $H^1$ clearly suffices for the variational energies to be well defined, one typically needs another derivative to perform gradient-based optimization schemes. For PINNs, the authors seem to already consider $\operatorname{ReLU}^3$ networks (i.e. the extra derivative), at least in the case of the results for Sobolev functions.

**Questions For Authors:**

- Can the results be extended to more general activation functions?

**Relation To Broader Scientific Literature:**

The work is well contextualized.

**Theoretical Claims:**

I checked the proof strategy but not the technical details.

---

> ### Author Rebuttal · Authors · 2025-03-31
>
> We are grateful for your thorough review and thoughtful suggestions. Below we provide detailed responses to each of your comments.
>
> **Q1**: Missing experiments.
>
> **A1**: We agree that numerical validation would enhance the work.  About experimental validation, the experiments in [1] have demonstrated that even in 100-dimensional cases, the Deep Ritz Method (DRM) attains a relative error of less than 20% for the Poisson equation and static Schrödinger equation in the setting of Barron space. Furthermore, when comparing experimental results with theoretical bounds, [1] observed that the convergence rates in their generalization bounds may not be sharp. Consequently, the experimental findings in [1] provide partial support for our results, as our theoretical framework aligns with theirs—while offering improved generalization bounds.
>
> Regarding why we did not include experiments: Like most theoretical papers, we have chosen to omit optimization error. In future work, we plan to account for optimization error and conduct experiments to validate our findings.
>
> **Q2**: Consider $H^2$ conforming networks for Deep Ritz Method (DRM) .
>
> **A2**: We totally agree with you. In fact, if we assume that the solutions belong to Barron space $\mathcal{B}^s(\Omega)$ with $s\geq 3$,  then $\text{ReLU}^2$ activation functions (i.e. $H^2$ conforming networks) could be employed for DRM, and our theoretical framework Theorem 2.4&2.5 still remains valid.
>
> In this work, both DRM and PINNs are analyzed under the weakest solution regularity assumption -- specifically, we assume the solutions reside in relatively low-order Barron spaces (e.g. $\mathcal{B}^3(\Omega)$ in Theorem 3.4). This fundamental assumption governs our choice of activation functions throughout the paper.  For example, for solutions in $\mathcal{B}^s(\Omega)$ with $s\geq 4$, $\text{ReLU}^3$ activations would become feasible for PINNs, and our theoretical framework Theorem 3.4 remains valid.
>
> **Q3**: Regarding the positioning of this work.
>
> **A3**: We can understand the reviewer’s concern about whether our purely theoretical error analysis fits ICML’s scope, especially given its focus on theoretical contributions without numerical simulations. However, **we believe this work aligns with ICML’s mission to advance  machine learning theory, as it addresses certain challenges arising in physics-informed learning where commonly used ML tools fail or are insufficient**. Whereas papers in mathematical journals often rely on numerical integration with well-established error estimates, our work tackles the more complex setting of Monte Carlo methods for computing high-dimensional integrals arising in neural network loss functions---a setting that requires different theoretical tools. Specifically, we contribute by (1) developing a new theoretical framework for physics-informed learning and (2) deriving meaningful error bounds for Barron spaces in the overparameterized regime, advancing the theoretical foundations of machine learning for scientific computing domain.
>
> **Q4**: Missing references.
>
> **A4**: We sincerely appreciate the reviewer for pointing out this important reference. It indeed provides valuable insights, particularly for the second-order elliptic equations where Lemma 1 parallels our Lemma C.11. Additionally, this reference  provides a thorough analysis for other types of PDEs, which broadens the scope. In the revised manuscript, we will cite this reference and incorporate this discussion on page 8 to offer readers a more comprehensive perspective.
>
> **Q5**: Extension to more general activation functions.
>
> **A5**: While our current analysis primarily focuses on $\text{ReLU}^k$
> activation functions, the framework can indeed be extended to more general activations. Here we outline a concrete example using tanh activations for the static Schrödinger equation (Theorem 2.5(2)):
>
> For the Deep Ritz method, the generalization bound under this setting takes the form:
> $$\frac{\alpha}{n}+\epsilon _{app}^2,$$
> where $n$ is the sample size, $\alpha$ represents the neural network complexity and $\epsilon _{app}$ is the approximation error in the $H^1$ norm. When employing tanh activations, Theorem 5.1 of [2] demonstrates that for the solution $u _{S}^{* }\in W^{k,\infty}$
> of the static Schrödinger equation, there exists a two-hidden-layer tanh network with width at most $\mathcal{O}(N^d)$ achieving an approximation error $\epsilon _{app}=\mathcal{O}(\frac{1}{N^{k-1}})$. In this setting, we have $\alpha=\mathcal{O}(N^d)$. Then taking a proper $N$ yields:
> $$\frac{\alpha}{n}+\epsilon _{app}^2=\frac{N^d}{n}+\frac{1}{N^{2(k-1)}}=\mathcal{O}(n^{-\frac{2k-2}{d+2k-2}}).$$
> This matches the convergence rate derived in Theorem 2.5(2) of our work.
>
> **References**:
>
> [1]: A Priori Generalization Analysis of the Deep Ritz Method for Solving High Dimensional Elliptic Partial Differential Equations. COLT 2021.
>
> [2]: On the approximation of functions by tanh neural networks. Neural Networks 2021.

---

### Official Review · Reviewer_iXkp · 2025-03-12

**Overall Recommendation:** 3

**Summary:**

This paper presents a refined generalization analysis of two popular deep learning-based methods for solving partial differential equations (PDEs): the Deep Ritz Method (DRM) and Physics-Informed Neural Networks (PINNs). The authors derive sharper generalization bounds for these methods under different assumptions about the solutions of the PDEs, particularly when the solutions lie in Barron spaces or Sobolev spaces.

**Claims And Evidence:**

Yes

**Essential References Not Discussed:**

The below paper is about the theory of PINNs by CNN.

Lei, G., Lei, Z., Shi, L., Zeng, C., & Zhou, D. X. (2025). Solving PDEs on spheres with physics-informed convolutional neural networks. Applied and Computational Harmonic Analysis, 74, 101714.

**Experimental Designs Or Analyses:**

No

**Methods And Evaluation Criteria:**

Yes

**Other Comments Or Suggestions:**

No

**Other Strengths And Weaknesses:**

Strengths:
1. The authors achieve sharper generalization bounds compared to previous works, particularly for DRM and PINNs, which is a solid contribution to the field.

2. The proposed methods provide a unified framework for deriving generalization bounds for machine learning-based PDE solvers, which can be extended to other PDEs and methods.

Weaknesses:
1. While the theoretical contributions are solid, the paper lacks empirical validation. It would be beneficial to see how the derived bounds hold in practice, especially for high-dimensional PDEs.

2. The analysis relies on the assumption that the solutions lie in Barron spaces or Sobolev spaces. While these are reasonable assumptions, they may not hold for all PDEs, and the paper does not discuss the implications when these assumptions are violated.

**Questions For Authors:**

1. The paper claims that the framework can be extended to other PDEs, including time-dependent ones. Could the authors provide some insights on how this extension would work?

2. How do the generalization bounds of DRM and PINNs compare with those of traditional numerical methods like finite element or finite difference methods, especially in high dimensions?

3. There may be some concerns on the use of $ReLU^k$ in deep neural networks. Are they commonly used in sovling PDE?

4. Do you have any insights regarding the optimality of the derived rate?

**Relation To Broader Scientific Literature:**

The contributions of the paper are related to the community of deep learning theory and scientific machine learning.

**Theoretical Claims:**

No

---

> ### Author Rebuttal · Authors · 2025-03-31
>
> Thank you for taking the time to review our article and for your insightful comments. Let us address your concerns point by point.
>
> **Q1**: Missing references.
>
> **A1**: We appreciate the reviewer for pointing out this important literature. It has established rigorous analysis of PICNN on the sphere, bridging the gap in understanding PINNs on manifolds. In the revised version, we will cite it with proper discussion.
>
> **Q2**: Experiments.
>
> **A2**: Regarding why we did not include experiments, like most theoretical papers, we have chosen to omit optimization error. In future work, we plan to account for optimization error and conduct experiments to validate our theory. For experimental validation of the Deep Ritz Method (DRM) for high-dimensional PDEs in the setting of Barron spaces, the experiments in [1] demonstrate that even in 100-dimensional cases, the DRM achieves a relative error of less than $20$% for Poisson equation and static Schrödinger equation. Moreover, when comparing experimental results with theoretical bounds, [1] observed that the convergence rates in their generalization error may not be sharp. Therefore, their experiments can, to some extent, support our findings, as our theoretical setting aligns with theirs—while we provide better generalization bounds.
>
> **Q3**: Assumptions for the solutions.
>
> **A3**: As the article primarily focuses on theoretical aspects, we–like most existing works–initially consider simpler cases. Investigating the setting where solutions belong to more complex spaces remains an important direction for future research.
>
> **Q4**: Extension to other PDEs.
>
> **A4**: In our analysis for PINNs, we require that the expected loss function of PINNs can be controlled by certain Sobolev norms between the exact and approximate solutions. For time-dependent PDEs, such as
> $$\partial_t u=\Delta u+f(u),$$
> where $u^{* }$ is the true solution and $f$ is the nonlinear term assumed to be Lipschitz. For simplicity we only consider the interior term, then the loss function is
> $$L(u)=||\partial _t u-\Delta u-f(u)|| _{L^2(0,T;L^2(\Omega))}^2.$$
> Then we have
> \begin{equation*}
> \begin{aligned}
> L(u)&=||\partial _t u-\Delta u-f(u)-\partial _t u^{* }+\Delta u^{* }+f(u^{* })|| _{L^2(0,T;L^2(\Omega))}^2\\\\
> &\lesssim ||\partial _t u-\partial _t u^{* }||  _{L^2(0,T;L^2(\Omega))}^2+||\Delta u-\Delta u^{* }|| _{L^2(0,T;L^2(\Omega))}^2+||u-u^{* } || _{L^2(0,T;L^2(\Omega))}^2.
> \end{aligned}
> \end{equation*}
> Then combining our method with neural network approximation results for spatiotemporal Sobolev spaces can yield similar generalization bounds.
>
> **Q5**: Comparison with traditional methods.
>
> **A5**: The error bounds of finite element method depend on the maximum mesh diameter $h$ and the computational complexity scales as $O(h^{-d})$ in high dimensions. Moreover, the error constants grow significantly with increasing dimension $d$, leading to the curse of dimensionality.
>
> In this work, under the assumption that the solutions belong to Barron spaces, the relevant constants demonstrate at most polynomial dependence on the dimension, making the generalization bounds meaningful even in high-dimensional settings.
>
> **Q6**: The use of $\text{ReLU}^k$.
>
> **A6**: The choice of activation functions is just to demonstrate that our method can achieve better results. Our method also works for other activations. To illustrate this, we provide a concrete example using tanh for the static Schrödinger equation.
>
> For the DRM, the generalization bound takes the form
> $$\frac{\alpha}{n}+\epsilon _{app}^2,$$
> where $n$ is the sample size, $\alpha$ represents the neural network complexity and $\epsilon _{app}$ is the approximation error. Theorem 5.1 of [2] shows that for the solution $u _{S}^{* }\in W^{k,\infty}$
> of the static Schrödinger equation, there exists a two-hidden-layer tanh network with width $O(N^d)$ achieving an approximation error $O(N^{-(k-1)})$. In this case, we have  $\alpha=O(N^d)$, then taking a proper $N$ yields
> $$\frac{\alpha}{n}+\epsilon _{app}^2=\frac{N^d}{n}+\frac{1}{N^{2(k-1)}}=O(n^{-\frac{2k-2}{d+2k-2}}).$$
> This result exactly coincides with the derived rate in Theorem 2.5(2) of our work.
>
> **Q7**: Optimality.
>
> **A7**: Some studies have shown that neural network-based estimators can achieve the minimax optimal rates for regression problems. These upper and lower bounds are estimated under the $L^2$ norm. However, for PINNs, different PDEs may require distinct norms  to measure the discrepancy between approximate and exact solutions, which significantly differs from the framework of regression problems. Therefore, whether the improved generalization bounds for PINNs derived in this work are indeed optimal still requires further validation in future studies.
>
> **References**:
>
> [1]: A Priori Generalization Analysis of the Deep Ritz Method for Solving
> High Dimensional Elliptic Partial Differential Equations. COLT 2021.
>
> [2]: On the approximation of functions by tanh neural networks. Neural Networks 2021.

---

### Official Review · Reviewer_sXtr · 2025-03-18

**Overall Recommendation:** 3

**Summary:**

The paper presents refined generalization error bounds for two Machine Learning (ML) based methods used to solve partial differential equations (PDEs) - the Deep Ritz Method (DRM) and Physics-Informed Neural Networks (PINNs). The main technical contribution made in this paper is to provide a sharper generalization bound for DRMs and PINNs based on localized Rademacher complexity techniques. For the DRMs, the analysis is mainly based on the Poisson Equation and the static Schrodinger Equation on the high-dimensional unit hypercube with Neumann boundary condition. For the PINNs, the paper focuses on the general linear second elliptic PDEs with Dirichlet boundary condition. Compared to previous studies, tighter bounds get obtained and unrealistic assumptions also get removed.

## Update After Rebuttal
The reviewer remains positive about the theoretical results presented in the paper, so the reviewer would like to remain the score. Nevertheless, the authors should probably discuss their answers to Q3 and Q4 mentioned in the review in detail when revising the manuscript and include some numerical experiments if possible.

**Claims And Evidence:**

The main claim made in this paper is that the convergence rates associated with the DRM and PINN can be further improved based on techniques like the peeling method and localized Rademacher complexity. Detailed proofs are provided to support the main claim here.

**Essential References Not Discussed:**

Given that solving PDEs via ML-based methods is a popular field recently, it might be beneficial for the authors to do a review of related methodology proposed in the current literature. See for instance the literature review section in [1]. Some important references that the authors didn't cite here include [2,3,4,5,6,7,8].

References:

[1] Lu, Y., Blanchet, J. and Ying, L., 2022. Sobolev acceleration and statistical optimality for learning elliptic equations via gradient descent. Advances in Neural Information Processing Systems, 35, pp.33233-33247.

[2] Li, Z., Kovachki, N., Azizzadenesheli, K., Liu, B., Bhattacharya, K., Stuart, A. and Anandkumar, A., 2020. Fourier neural operator for parametric partial differential equations. arXiv preprint arXiv:2010.08895.

[3] Lu, L., Jin, P., Pang, G., Zhang, Z. and Karniadakis, G.E., 2021. Learning nonlinear operators via DeepONet based on the universal approximation theorem of operators. Nature machine intelligence, 3(3), pp.218-229.

[4] Chen, Y., Hosseini, B., Owhadi, H. and Stuart, A.M., 2021. Solving and learning nonlinear PDEs with Gaussian processes. Journal of Computational Physics, 447, p.110668.

[5] Han, J., Jentzen, A. and E, W., 2018. Solving high-dimensional partial differential equations using deep learning. Proceedings of the National Academy of Sciences, 115(34), pp.8505-8510.

[6] Khoo, Y., Lu, J. and Ying, L., 2021. Solving parametric PDE problems with artificial neural networks. European Journal of Applied Mathematics, 32(3), pp.421-435.

[7] Sirignano, J. and Spiliopoulos, K., 2018. DGM: A deep learning algorithm for solving partial differential equations. Journal of computational physics, 375, pp.1339-1364.

[8] Zang, Y., Bao, G., Ye, X. and Zhou, H., 2020. Weak adversarial networks for high-dimensional partial differential equations. Journal of Computational Physics, 411, p.109409.

**Experimental Designs Or Analyses:**

N/A (This is a theoretical paper). However, the reviewer does think that the paper's impact might be increased if the authors can add a few numerical experiments to justify their findings.

**Methods And Evaluation Criteria:**

N/A (This is a theoretical paper). However, the reviewer does think that the paper's impact might be increased if the authors can add a few numerical experiments to justify their findings.

**Other Comments Or Suggestions:**

N/A

**Other Strengths And Weaknesses:**

The main strength of this article is the techniques used in its proofs, which does address issues like the infeasible assumption (some solution represented by a neural network in the $H^{1}_0$ space). However, for the weakness of this article, the reviewer's main concern is that the authors might have to compare their work with the previous work [1] in a more thorough way. Firstly, given that [1] establishes not only upper bounds but also information theoretical lower bounds on the expected estimation error, would it be possible for the authors to provide some intuition on whether it would be possible to derive information theoretic lower bounds for the cases considered in this paper? (i.e, are the bounds established here minimax optimal or not?) To the best of the reviewer's knowledge, this seems to be one important criteria for articles focusing on learning theory and nonparametric statistics. Secondly, it seems that [1] also used techniques like peeling and localized Rademacher complexity, which is similar to the proof strategy deployed in this paper - would it be possible for the authors to further comment on the proof novelty in this paper?

References:

[1] Lu, Y., Chen, H., Lu, J., Ying, L. and Blanchet, J., 2021. Machine learning for elliptic pdes: Fast rate generalization bound, neural scaling law and minimax optimality. In International Conference on Learning Representations, 2022. URL https://openreview.net/forum?id=mhYUBYNoGz.

**Questions For Authors:**

N/A

**Relation To Broader Scientific Literature:**

This paper, which studies how to improve the convergence rate of the DRMs and PINNs, should be mainly situated as the application of nonparametric statistics and learning theory (high-dimensional statistics) in scientific machine learning (SciML/AI4Science), i.e., the theoretical analysis of algorithms in SciML/AI4Science.

**Theoretical Claims:**

For DRMs the main theoretical claims are in section2, which are results for the Poisson Equation and the static Schrodinger Equation on the high-dimensional unit hypercube with Neumann boundary condition. For PINNs the main theoretical claims are in section 3, which are results for general linear second elliptic PDEs with Dirichlet boundary condition. The reviewer didn't find any significant issue with the proofs of the main claims.

---

> ### Author Rebuttal · Authors · 2025-03-31
>
> We sincerely appreciate your time in reviewing our manuscript and your valuable insights. Below we provide a point-by-point response to your concerns.
>
> **Q1**: Experiments.
>
> **A1**: We agree that experiments would further strengthen our work. In future work, we plan to account for optimization error and conduct experiments to validate our results. For experimental validation of DRM in the setting of Barron space, the experiments in [4] show that even in 100-dimensional cases, the DRM achieves a relative error of less than $20$%. Moreover, when comparing experimental results with theoretical bounds, [4] found that the convergence rates in their generalization error may not be sharp. Therefore, their experiments can, to some extent, justify our findings, as our theoretical setting aligns with theirs—but we provide better generalization bounds.
>
> **Q2**: References.
>
> **A2**: We sincerely appreciate the reviewer for pointing out these important references. In the revised version, we will follow the format of [1] and incorporate all suggested references [2-8] with proper discussion.
>
> **Q3**: Lower bounds.
>
> **A3**: What [1] has achieved better than our work is that they also derived lower bounds for both DRM and PINNs. Their results show that the bound for DRM is not minimax optimal, whereas that for PINNs is minimax optimal. However, the metric used in [1] to evaluate PINNs is the $H^2$ norm, which requires strong convexity assumption on PDEs and neural network functions to belong to $H_0^1$. Such assumptions appear too stringent.
>
> Recent studies have shown that neural network-based estimators can achieve minimax optimal rates for regression problems. These bounds are estimated under the $L^2$ norm. However, for PINNs, different PDEs  require distinct norms to measure the discrepancy between the empirical and true solutions, which differs significantly from the regression framework. Therefore, whether the bounds we derived for PINNs and DRM are truly optimal still requires further investigation in future research.
>
> **Q4**: Comparison of proof strategies with [1].
>
> **A4**: For the DRM, we also analyze the Barron space setting and derive novel generalization bounds that remain meaningful even in overparameterized regimes. This advancement surpasses the capabilities of [1]'s approach (see Proposition 2.7). For the Poisson equation, the variational form $\mathcal{E} _{P}(u)$ is not equal to the expectation of its corresponding empirical part $\mathcal{E} _{n,P}(u)$. This makes the local Rademacher complexity (LRC, [3]) and the method in [1] infeasible. Morover, for example, the core lemma (Lemma B.4) in [1] requires the condition (from [3]) that
> $$R _n({f\in \mathcal{F}: \mathbb{E}[f]\leq r}) \leq \phi(r),$$
> where $R _n$ is the empirical Rademacher complexity. Then in [1], an appropriate functions class $\mathcal{F}$ can be chosen such that $\mathbb{E}[f] = \mathcal{E} _{P}(u)-\mathcal{E} _{P}(u^{* })$. By doing so, the strong convexity can be used. However, for the Poisson equation, there does not exist a function class $\mathcal{F}$ such that $\mathbb{E}[f]=\mathcal{E} _{P}(u)-\mathcal{E} _{P}(u^{* })$. In contrast, this work develops a novel peeling technique to establish fast rates. The static Schrödinger equation shares similar strong convexity properties with certain classical problems like bounded-noise regression [2]. Although approaches from [1], [2] and our Poisson equation method remain applicable, they are much complicated. To address this, we instead develop a novel error decomposition method that allows direct application of the results in [3] (LRC), yielding better generalization bounds through a more concise proof framework.
>
> For PINNs, [1] only considered the case of the static Schrödinger equation and assumed that the neural network function class is a subset of $H_0^1$. This assumption makes PINNs contain only interior terms, thereby giving them a strong convexity property similar to that of the DRM. Thus, the approach for PINNs in [1] is identical to that for the DRM. In contrast, we consider more general PDEs and treat PINNs as a multi-task learning (MTL) problem. The key difference from the DRM is that for PINNs, we only require a non-exact form of oracle inequality, eliminating the need for the strong convexity. Then, by using LRC under MTL (where we also derive an improved Talagrand-type concentration inequality for MTL), we obtain sharper generalization bounds. Moreover, this approach can also be extended to other types of PDEs and neural-network-based PDE-solving methods similar to PINNs.
>
> **References**:
>
> [1] Machine learning for elliptic pdes: Fast rate generalization bound, neural scaling law and minimax optimality. ICLR,2022.
>
> [2] Deep neural networks for estimation and inference. Econometrica,2021.
>
> [3] Local Rademacher complexities. AoS,2005.
>
> [4] A Priori Generalization Analysis of the Deep Ritz Method for Solving
> High Dimensional Elliptic Partial Differential Equations. COLT,2021.

---

> > ### Comment · Reviewer_sXtr · 2025-04-06
> >
> > The reviewer would like to thank the authors for addressing the two main concerns. The reviewer remains positive about the theoretical results and will keep the score. However, the authors are encouraged to discuss their answers to Q3 and Q4 above in detail in the revised version of the manuscript and include some numerical experiments if possible.

---

> > > ### Author Response · Authors · 2025-04-06
> > >
> > > We sincerely appreciate your constructive feedback. In the revised manuscript, we will expand the discussion of Q3 and Q4 with additional analysis and incorporate numerical experiments to support our theoretical results. Thank you for your valuable suggestions.

---

### Decision · Program_Chairs · 2025-05-01

**Decision:**

Accept (poster)

**Comment:**

The manuscript considers generalization error analysis for two approaches of using neural networks to solve PDEs. The reviewers are unanimously positive and the authors have addressed most concerns during the discussion phase.